# On the Continuum of Foundational Validity: Lessons from Eyewitness Science for Latent Fingerprint Examination

**DOI:** 10.3390/bs15091145

**Published:** 2025-08-22

**Authors:** Adele Quigley-McBride, T. L. Blackall

**Affiliations:** Department of Psychology, Simon Fraser University, Burnaby, BC V5A 1S6, Canada; tlb14@sfu.ca

**Keywords:** latent print examination, eyewitness identification, forensic science, evidence, foundational validity

## Abstract

Whether forensic disciplines have established foundational validity—sufficient empirical evidence that a method reliably produces a predictable level of performance—has become a question of growing interest among scientists and legal professionals. This paper evaluates the foundational validity of two sources of forensic evidence relied upon in criminal cases: eyewitness identification decisions and latent fingerprint examiners’ conclusions. Importantly, establishing foundational validity and estimating accuracy are conceptually and functionally different. Though eyewitnesses can often be mistaken, identification procedures recommended by researchers are grounded in decades of programmatic research that justifies the use of methods that improve the reliability of eyewitness decisions. In contrast, latent print research suggests that expert examiners can be very accurate, but foundational validity in this field is limited by an overreliance on a handful of black-box studies, the dismissal of smaller-scale, yet high-quality, research, and a tendency to treat foundational validity as a fixed destination rather than a continuum. Critically, the lack of a standardized method means that any estimates of examiner performance are not tied to any specific approach to latent print examination. Despite promising early work, until the field adopts and tests well-defined procedures, foundational validity in latent print examination will remain a goal still to be achieved.

## 1. On the Continuum of Foundational Validity: Lessons from Eyewitness Science for Latent Fingerprint Examination

After decades of a comfortable reputation among investigators, legal professionals, and lay persons alike, forensic science faced increasing scrutiny in the 1990s and 2000s. The 2009 National Research Council report ([46]) is considered by many researchers and practitioners to be a turning point, as it clearly laid out several serious shortcomings in forensic science that had previously escaped formal scrutiny. The report was especially critical of *pattern-matching disciplines*—fields in which practitioners visually compare patterns or markings between two samples (e.g., latent fingerprint examination). In these disciplines, experts determine whether two pieces of evidence are sufficiently similar to conclude they share a common source (e.g., a suspect). Many researchers and practitioners acknowledged that the report exposed fundamental weaknesses in the scientific foundations of many forensic disciplines, prompting a wave of new research to address these concerns in forensic science.

There is now substantially more high-quality foundational research that can speak to reliability and the likelihood of error across a range of forensic disciplines. As a result, forensic service providers (FSPs) and legal professionals who use the results produced by forensic laboratories have started to ask how much research is enough. In particular, when considering court admissibility requirements for this type of information, the test in [15] ([15]) requires “general acceptance in the relevant field”, the test from [9] ([9]) requires knowledge of error rates, and the recently updated [14] ([14]) requires “reliable principles and methods”, generally in that discipline and as applied in that case. Thus, whether a forensic discipline has sufficient foundational research can, in theory, determine the extent to which that type of evidence can be relied upon in court cases.

In 2016, the President’s Council of Advisors on Science and Technology ([49]) published a review of several forensic disciplines—single-source DNA, DNA mixtures, bitemarks, latent fingerprints, firearms, footwear, and hair analysis. Their mandate was to determine whether these disciplines had demonstrated *foundational validity*, defined as the extent to which a method has been empirically shown to produce accurate and consistent results, based on peer-reviewed, published studies ([49]). Specifically, PCAST evaluated whether each discipline’s procedures had been tested for repeatability (within examiner), reproducibility (across examiner), and accuracy under conditions representative of actual casework (see also the discussion in [58]).

Importantly, PCAST emphasized that foundational validity is a property of the specific method under consideration, rather than a property of performance outcomes. That is, a discipline can lack foundational validity even when examiners achieve accurate results provided that their success cannot be attributed to a clearly defined and consistently applied method that can be independently replicated (see also [80]). Given these implications, it is unsurprising that methodological precision and compliance have received increasing attention in recent years ([60]). Without a clear and consistently applied method, results from studies designed to observe performance metrics reflect the accuracy achieved by an undefined mix of examiner strategies that cannot be meaningfully linked to any particular approach and are, consequently, difficult to interpret, predict, or replicate.

Of the disciplines reviewed, only single-source DNA, DNA mixtures with no more than three contributors, and *latent print examination* (LPE) passed the muster. However, although the PCAST report cited other studies, this declaration about LPE was based on a total of two black-box studies[note 1] ([48]; [62]), only one of which (Ulery et al.) had been published in a peer-reviewed journal. PCAST concluded that LPE could be considered foundationally valid if its limitations were clearly communicated when presenting results. Since the report, one additional LPE black-box study has been published ([23]), resembling those cited by PCAST. If black-box studies are indeed the primary evidence required to establish foundational validity, as the report implies, then the field remains in a similar position nearly a decade later.

So, is the existing body of empirical research on LPE sufficient to establish its foundational validity? In experimental psychology, most researchers would likely agree that three studies conducted under a narrow set of conditions may be promising and worthy of further exploration, but they are insufficient for making broad policy recommendations about the practices being evaluated. At the same time, researchers in psychology would not necessarily dismiss studies that fall outside of the strict black-box criteria outlined by [49] ([49]); a diverse range of empirical work can offer valuable insights. These considerations raise important questions: How much research is enough to consider a method foundationally valid? At what point can the latent fingerprint community be confident that their methods yield consistent results—both between (reproducible) and within examiners (repeatable; [58])?

One way to answer this question is to look at other scientific fields that rely on expert perceptual judgments, such as radiology. Like forensic examiners, radiologists make judgments based on their perception and interpretation of complex, visually presented data to identify medical issues. However, radiology and other fields operating within medical systems benefit from methods that are well standardized, institutional safeguards designed to catch and correct errors, and formal oversight mechanisms. In contrast, many forensic disciplines, including LPE, operate under loosely defined frameworks and often lack the systemic infrastructure seen in fields like radiology. These differences, considered alongside the limited body of rigorous scientific research, make comparisons between forensic and medical disciplines difficult and potentially unproductive.

A better comparison might be found in the collection and use of a different type of forensic evidence: *eyewitness identification*. Like forensic pattern-matching disciplines, eyewitness evidence has faced, and continues to face, significant scrutiny over its reliability. Eyewitness evidence depends heavily on human perception and judgment, both on the part of the eyewitness and the investigators handling the case. Eyewitness data are typically collected under circumstances where decision-makers have wide discretion but are also often lacking the resources and personnel needed to adhere to research-based best practices ([73]). Both types of evidence have a strong influence on end-users—a confident eyewitness is highly persuasive ([17]; [39]), much like a forensic expert is when they are testifying to their results during a trial ([7]; [8]). Many eyewitness identification studies focus on developing procedures that preserve the eyewitness’s memory for the suspect and appropriately test the extent to which the police suspect matches the eyewitness’s memory. The ultimate goal, as in pattern-matching disciplines, is to link a specific individual to a crime event.

Yet, eyewitnesses are mistaken at an alarming rate. Even when identification procedures follow best practices, approximately one-third of real eyewitnesses identify a known-innocent person (an intentionally placed *filler*; Wells et al., 2015). So, although research shows that eyewitnesses are less accurate than LPEs, there is a robust body of empirical research supporting the methods recommended for use in practice, which is important for foundational validity. This makes eyewitness identification a compelling example of how foundational validity can be achieved even in the face of known performance limitations.

There are also clear differences between eyewitness identification and LPE (see Table 1). For instance, eyewitness identification science examines a memory-based task rather than a perceptual comparison task. However, the cognitive processes and legal actors involved in establishing and using eyewitness evidence and forensic pattern-matching disciplines are remarkably similar ([51]). Standardized eyewitness identification procedures are also supported by decades of research, beginning in the 1970s, clarifying how and why various factors affect eyewitness reliability.

For instance, there are multiple studies demonstrating that the quality of the initial memory will determine whether the eyewitness is able to make a correct identification later and how that is related to their certainty about that identification decision ([81]). This is akin to the impact of the quality of the trace evidence collected from the crime scene, as this will limit what conclusions can be made by LPEs and the associated certainty ([12]). Research also shows that the methods used to collect eyewitness accounts can change both an eyewitness’s willingness to make an identification, their accuracy, and their confidence in their decision ([73]). This can be likened to the methods and standard operating procedures maintained by FSPs used to collect, examine, and evaluate forensic evidence ([60]). Finally, any additional information the witness receives between the crime and the recalled event can affect their accuracy and confidence ([40]), similar to the effect of various types of potentially biasing information on forensic examiners ([27]).

In this paper, eyewitness identification science is presented as an example of a programmatic, rigorous body of cumulative science that sheds light on the strengths and weaknesses of different approaches to gathering forensically-relevant evidence. To be clear, though, the goal is not to present eyewitness science as the standard against which to compare forensic disciplines to determine foundational validity. Nor do we dispute studies that show low error rates among LPEs, suggesting that they can accurately sort between same- and different-source latent–exemplar pairs under the conditions that have been tested (e.g., [62]; [23]). As noted, however, accuracy is not synonymous with foundational validity.

Foundational validity is not a “destination”. Forensic disciplines lie on a continuum from no foundational validity through to the aspirational goal of understanding everything about how and why a method or technique produces favorable outcomes under some circumstances and not in others. Eyewitness science has come a long way along this continuum, but the journey has not always been linear, and there is still much to test and understand. In this article, we compare and contrast the current state of eyewitness identification science with that of latent print examination—not because LPE lacks scientific grounding but because, aside from DNA analysis, it is the only other forensic discipline that has been recognized as possessing some degree of foundational validity by both practitioners and scientists ([49]).

## 2. Where Does Eyewitness Science Lie on the Foundational Validity Continuum?

Eyewitness identification evidence is frequently used in the criminal legal system and has a broad body of research guiding best practices. Of course, this was not always the case. Eyewitness testimony also has a much longer history as a source of evidence, perhaps even since the very first person accused another of a misdeed, than physical forensic evidence, such as fingerprints. The emergence of systematic programs of research on eyewitness memory and identification procedures began in the 1960s and 1970s but really started to surge in the 1980s and 1990s ([76]).

During this time, foundational work established what are now considered highly replicable, basic findings. For instance, early work by Professor Elizabeth Loftus and colleagues demonstrated how easy it was for information encountered after a crime event to alter memory, often without the eyewitness realizing it ([41]), and, later, that entirely false memories could be implanted ([42]). In the 1970s, Professor Gary Wells introduced a key conceptual distinction between *system variables*—factors that could be controlled by police and other agents in the legal system—and *estimator variables*—factors associated with the crime event, culprit, or witness that cannot be controlled, guiding applied research in this area ([71]). Wells and his colleagues demonstrated that the use of properly administered lineup procedures—system variables—was a reliable way for police to ensure an inaccurate eyewitnesses did not derail an investigation or result in the conviction of the wrong person. Throughout the 1970s, 1980s, and 1990s, eyewitness researchers ran studies to clarify best practices surrounding the creation and administration of lineups to ensure these procedures reduced the risk to innocent suspects while maintaining the ability to obtain an identification of the correct person as much as possible (e.g., [74], [75], [77]). By the late 1990s, eyewitness researchers had built up a robust collection of findings that had been replicated many times.

Representing a culmination of this work, the American Psychology-Law Society (AP-LS) produced a Scientific Review Paper (SRP) in 1998 ([78]). This document outlined four scientifically grounded recommendations that had been well-studied and replicated. At the time, there was a broad consensus among eyewitness identification researchers that each of these recommendations had sufficient empirical support to justify its use in real cases. Even if future research found reasons to refine these recommendations, the points in the 1998 SRP could be relied upon. Figure 1 shows the types of citations used to justify the recommendations in 1998. Of the 152 total citations, 59 were controlled, experimental studies, 30 were review papers drawing on the wider literature, and five were meta-analyses that combined across studies to more accurately estimate the effect of variables or procedures. While this reference list is not a comprehensive list of all studies conducted prior to 1998, it represented what experts in the discipline considered to be key contributions to the science.

Although there was enough work to make concrete recommendations for testing eyewitness memory in real cases, this area continued to grow post-1998. In fact, the area grew so much that in 2020, another SRP was released by the AP-LS. The 2020 SRP includes nine recommendations to aid in the collection, preservation, and evaluation of eyewitness evidence in the field, and for many common real-world situations, researchers can now draw on several studies that can help evaluate the reliability of eyewitness evidence under similar circumstances. The 2020 SRP ([73]) was supported by more than 200 citations, including 145 experimental studies—115 of which were published after the 1998 SRP (see Figure 2). The report also drew on a larger number of meta-analyses (13 total, 11 post-1998), review papers (33 total, 29 post-1998), and other empirical and non-empirical publications (87 total, with 54 published after 1998; see Figure 2). Of course, the raw number of studies cited in a review paper is a crude measure of scientific progress. However, these citations were selected by experts to support formal policy recommendations and vetted through several rounds of comments, suggesting they meet a standard of quality broadly accepted by researchers in the field.

As with any scientific discipline, not all avenues pursued by eyewitness researchers have withstood the test of time. Some effects—particularly those associated with estimator variables like lighting, exposure duration, weapon focus, and cross-race identification—have proven robust in laboratory settings where such factors can be carefully controlled. However, these same factors can be difficult to assess in real cases, as documentation is sparse and often relies on subjective eyewitness recollections, resulting in mixed success replicating these effects in archival and real-world samples ([25]). Even findings once thought to be well established, such as own-race bias ([44]), have shown greater variability in more recent research ([36]).

Several system variable recommendations have also been revised or withdrawn in light of new evidence and do not appear in the 2020 SRP. For example, lineup administrators were once encouraged to include an “appearance change instruction,” warning witnesses that some of the culprit’s features may have changed since the crime ([5]). However, later research showed that this instruction increased eyewitnesses’ willingness to choose even when they were uncertain, increasing false identifications without boosting correct decisions ([45]). Similarly, sequential lineups—where photos are shown one at a time—were once preferred over simultaneous presentation ([38]). But a large field study found that, when other best practices were followed, there were only minimal differences in outcomes observed when lineups were presented sequentially versus simultaneously ([79]). Despite this, sequential lineups remain widely used in U.S. police departments ([18]), likely due to the broad success of disseminating this research in the past. Of course, it is clearly preferable to avoid retracted recommendations, but these examples demonstrate the self-correcting nature of science.

As a result of the body of research amassed by eyewitness researchers, there have been significant efforts to standardize the procedures used in the legal system by disseminating research-based recommendations to police and other policymakers. For example, the 2020 SRP provides guidance to police at every stage of the investigation. There are recommendations regarding the collection of eyewitness accounts after a crime event, how best to preserve eyewitness memory between the crime event and any subsequent recall session or identification procedure (e.g., asking witnesses not to discuss the event or search social media), how best to conduct an identification procedure (e.g., using a fair lineup with one suspect and five known-innocent fillers, administered by someone who does not know the suspect’s identity; ([73])), and observations that can indicate whether an eyewitness’s identification was accurate or not (reflector variables; [72]). Thus, eyewitness science has developed programmatic research identifying specific, empirically supported methods for collecting and evaluating eyewitness evidence—methods shown to improve identification outcomes.

Since the 1998 SRP, there have been several field studies demonstrating the benefits of the recommended procedures with real eyewitnesses in real cases. These studies demonstrate the number of filler identifications (decisions were known to be inaccurate in real cases; [79]), the effect of a biased lineup ([57]), and indicators of eyewitness accuracy ([52]; [55]) in real cases. As a result, the evidence supporting these procedures is no longer limited to laboratory contexts, strengthening the credibility of this guidance. Of course, changes to police policies and common law take time, and standardization of methods remains inconsistent across agencies and jurisdictions in the USA ([18]). There will also always be real-world cases that are so unusual or unique that there is no existing research relevant to those case circumstances. Some real-world cases are also so unusual that no existing research is directly relevant. Still, so long as an agency documents the identification procedures used, there is ample empirical research addressing when and why eyewitness evidence is likely to be reliable—or not.

Though more research will always be necessary, eyewitness science has made substantial progress in identifying methods that produce reliable outcomes and using that research to bring about policy change. The nine recommendations made in the 2020 SRP have reached a level of foundational validity that almost every eyewitness researcher agrees with ([56]). Other recommendations enjoyed similar levels of empirical support and expert consensus but fell just short of the threshold for inclusion in the 2020 SRP—placing them slightly lower on the continuum of foundational validity.

Thus, eyewitness researchers have amassed enough evidence to recommend specific procedures and methods to preserve eyewitness evidence. Still, there are many areas of eyewitness work where researchers’ understanding is still limited or developing, such as the study of reflector variables ([72]). These are observations that can be made during or immediately after a lineup that change in predictable ways when an eyewitness is correct versus incorrect in any particular case. Foundational validity, or how best to collect and use this information, is still progressing in this area, though. One example is an eyewitness’s confidence in their decision—a highly confident eyewitness is more likely to be accurate than one who lacks confidence. However, researchers continue to debate how best to elicit, record, and interpret confidence and how to analyze it in experimental studies ([81]). So, foundational validity can vary across topics even within a single field. Scientific progress requires continuous empirical testing—both for established procedures and for those still gaining empirical support.

## 3. Where Does Latent Fingerprint Examination Lie on the Foundational Validity Continuum?

Latent fingerprint examination (LPE), a well-known and generally accepted forensic pattern-matching discipline, has made significant strides in creating a body of programmatic research to provide a scientific foundation to the discipline. In fact, several topics relevant to LPE are supported by a growing body of peer-reviewed publications. These include, but are not limited to, the following:Evaluations of techniques used to lift and process latent fingerprints at crime scenes (e.g., a review by [11]);Studies examining the use of algorithms or databases to support LPE (e.g., image quality assessment; [70]);Statistical models that quantify the value of any LPE results ([47]; [59]);Textbooks and training materials that are used to introduce trainees to the field (e.g., [2]; [24]).

In addition to peer-reviewed studies, considerable work has been developed outside traditional scientific channels—such as government reports, process maps, and consensus-based documents. Much of this growth has occurred since the 2009 NRC report.

While these developments contribute to the broader foundation of LPE, they are less relevant to the specific focus of this paper: the foundational validity of the human decision-making process in LPE. Just as the foundational validity of eyewitness identification depends on the methods used to collect and interpret witness memory, our focus is on the validity of the methods used by LPEs and the associated outcomes. This is related to, but distinct from, the question of LPE accuracy since there could be multiple foundationally valid methods that produce different levels of reliability and accuracy among LPEs that use them.

Controlled studies directly examining LPE judgments are fairly limited, with only a handful of black-box type studies (e.g., [3]; [23]; [31]; [48]; [62]), a small body of other work examining specific questions about examiner performance (e.g., [34]; [47]; [61]), and blind-proficiency testing efforts ([16]). These studies report mixed findings. Some suggest low false positive error rates (<1%; [62]; [23]) while others report higher rates of error (15.9% and 18.1%; [31]), depending on the conditions and materials tested.

Although fingerprint examination has better empirical support than many other forensic disciplines, is the existing body of research sufficient to establish the foundational validity of LPE judgments? Some researchers, practitioners, and government entities have suggested that it is. For instance, as noted here in the introduction, the PCAST report (2016) concluded that “latent fingerprint analysis is a foundationally valid subjective methodology—albeit with a false positive rate that is substantial and likely to be higher than expected by many jurors based on longstanding claims about the infallibility of fingerprint analysis” (p. 9). However, this conclusion was based on “only two properly designed studies of the foundational validity and accuracy of latent fingerprint analysis” (p. 101), as other studies measuring error rates among LPEs failed to meet PCAST’s criteria for methodological rigor. Although it is standard in science to interpret results cautiously based on study limitations, the studies dismissed by PCAST are not without value. In fact, subsequent peer-reviewed publications referring to the report have critiqued PCAST’s treatment of the LPE literature and the question of foundational validity (e.g., [23]; [31]).

Though limited, the existing research provides valuable insights into LPEs’ abilities. Under optimal conditions—such as when LPEs are not rushed, fatigued, stressed, or exposed to task-irrelevant information—false positive rates are generally below 1%, although false negative rates tend to be higher (e.g., 4.2% in [23]; 7.5% in [62]). Furthermore, exclusion decisions appear to involve qualitatively different decision processes than identification decisions ([68]). LPEs find fingerprint comparisons more difficult when the latent fingerprint is of poor quality (e.g., smudged or missing information), and they are more likely to find such prints to lack value for comparison ([64], [65]).

Expert LPEs consistently outperform laypersons, especially when the latent and the exemplar fingerprints appear highly similar but are not from the same source (a “*close non-match*”; [61]). However, these close non-matches are also associated with elevated error rates among expert LPEs ([31]). Research shows that LPEs can expect to examine close non-matches increasingly often due to improvements in database search algorithms that draw on databases with many potential candidates, increasing the chance that a very similar fingerprint from a different person is identified as a candidate ([4]; [37]).

Studies also demonstrate that LPE relies heavily on subjective human judgment, evidenced by both intra- and inter-examiner inconsistencies ([64], [65], [66], [67]). Emerging evidence suggests that individual differences in pattern-matching abilities might be trait-like, with some people naturally more adept at these tasks than others ([19], [21]). Finally, forensic examiners may behave more conservatively (i.e., less likely to make same-source judgment) when they know they are being tested, which is consistent with similar findings in other disciplines and is referred to as the Hawthorne effect ([43]; [53]; [54]).

Despite these advances, important limitations and knowledge gaps remain. Many findings have not been replicated across a variety of LPE samples—a significant portion of the error-rate literature originates from a single research group working with a dataset collected through a Noblis–FBI partnership (e.g., [23]; [62], [63], [64], [65], [66], [67]). While these studies are high-quality and influential, scientific fields benefit from research conducted by a diverse range of research teams with different approaches and perspectives. Moreover, existing error rate studies tend to rely on fingerprint impressions that are already processed to some degree, higher quality than latent fingerprints lifted from crime scenes, and examined under conditions where the LPEs know that they are being tested by researchers who will compare their judgments to the correct answers. There are also questions about whether the rate of same and different source latent–exemplar pairs in studies matches what occurs in regular casework. These are important deviations from casework that are known to impact how people make decisions—generally and specifically in LPE ([20]; [22]; [54]; see also critiques of fingerprint proficiency testing in ([30]); [32]). Thus, the error rates in operational settings may be different than those reported in the literature—particularly when considering the prevalence of close non-matches ([31]) and the influence of task-irrelevant, biasing contextual information ([12]; [50]) in real cases, which tend to change decision-making patterns.

Even if there were more error rate studies to draw on, though, we argue that concluding that LPE is foundationally valid would still be premature. Though the literature demonstrates that LPEs are able to distinguish between same- and different-source latent–exemplar fingerprint pairs at a much higher rate than novices and lay individuals, determining accuracy levels is different from determining whether a discipline is foundationally valid. Therefore, the most significant barrier to advancing foundational validity in LPE is the lack of a consistent methodological approach to examinations. Without consensus on a sufficiently detailed set of procedures and criteria—or even several clearly defined approaches—foundational validity cannot be established.

This is not to say that there is no consistency at all among LPEs. Most LPEs are trained to follow a procedural framework known as “ACE-V”, which outlines an order of operations. The examiner first Analyzes the latent fingerprint obtained from the crime scene to determine if there is a sufficient number of usable features such that the impression could be reasonably compared with another impression. Next, if the latent fingerprint is deemed to have analytic value, the examiner will perform a visual Comparison between the latent fingerprint and a fingerprint from a suspect or database search. Subsequently, the LPE will Evaluate the data and discern whether it is their opinion that these impressions are from the same source (“identification”), a different source (“exclusion”), or that there is not enough information to decide either way (“inconclusive”). Finally, that decision will be Verified by another examiner ([35]; [69]).

Although widely adopted, ACE-V prescribes only a general sequence of steps but does not standardize techniques or decision thresholds, leaving considerable room for examiner discretion. For example, ACE-V does not define how many or what types of features are required for comparison or identification, despite wide variability in these judgments ([64], [65]). Such subjectivity can lead to inconsistencies both across examiners (reproducibility) and within the same examiner under different conditions (repeatability; [63]). Moreover, ACE-V includes a “Verification” step, but does not specify how often verification should occur and for what types of decisions, or how that verification procedure should look. In particular, there is a lack of guidance on the importance of using a blind verifier—another examiner that does not know the original examiner’s conclusions. The difference in the potential for error with blind versus open verification methods, broadly defined, could be significant ([33]), but both are acceptable when using ACE-V ([69]) according to existing standards (e.g., ANSI/ASB BPR 144, ([1])).

Consequently, the ACE-V process, in its current form, does not ensure uniformity in decision-making or provide sufficient safeguards against examiner error. This flexibility also means that error rates reported in black box studies are not anchored to any standardized procedure and examiners participating in these studies may have employed vastly different internal criteria, despite ostensibly operating “within” the ACE-V framework. The literature currently suggests that false positive error rates could be less than 1% under some conditions and greater than 18% under others. Yet, it remains unclear how much of this variation is attributable to differences in the procedures and decision criteria used by each LPE.

In a recent publication, Swofford and colleagues (2024) highlighted the important distinction between *method performance* and *method conformance* in the development of forensic disciplines. Once a standardized method is implemented among LPEs, the performance of that method can be assessed in terms of reproducibility, repeatability, and false positive and negative error rates. The question of decision outcomes and performance, however, is separate from examining whether a given LPE in any particular case adhered to that standardized method. If an examiner deviates from the prescribed procedure, any resulting error may reflect non-conformance rather than a flaw in the method itself. Accordingly, empirical efforts should assess not only the reliability of standardized methods under controlled conditions but also the effects of deviations from those methods on the accuracy and defensibility of forensic conclusions.

This issue is not unique to fingerprint examination; it is a pervasive, systemic issue across forensic sciences. In the United States, even when consensus-based standards and best practice recommendations are published by organizations, such as the Academy Standards Board (ASB) or the Organization of Scientific Area Committees (OSAC), adherence is not mandatory, and oversight mechanisms are minimal or nonexistent. As a result, an FSP may claim to have adopted a standard, but there is no guarantee that it is being consistently followed in practice. Consequently, the protocols used by LPEs in the USA often vary by jurisdiction, laboratory, and even among individual LPEs—sometimes substantially. This reduces the applicability of the few existing studies on error rates among LPEs. Without knowing which methods, criteria, or procedures were used by the LPEs who participated in these studies, it remains an open question whether their reported performance metrics generalize to the method used in any given case.

## 4. Conclusions

This comparative analysis demonstrates that latent fingerprint examination currently occupies a lower position on the foundational validity continuum than eyewitness identification science. We arrive at this conclusion based on two overarching factors: (1) the current state of the empirical literature and (2) the lack of a standardized methodology in LPE.

### 4.1. What Is Known About Latent Fingerprint Examination Is Based on a Nascent Empirical Literature

A key difference between the two disciplines examined here is the breadth, depth, and strength of their respective evidence bases. When discussing the reliability of LPE, researchers, practitioners, and policymakers typically rely on just a few studies that meet the criteria for a “black-box” study (e.g., [48]; [62]; [23]). However, the performance of eyewitnesses during identification procedures can be estimated by referencing many studies—over 200 cited in the 2020 SRP—conducted by a variety of research groups. Individual studies might vary in methodological rigor or realism, but the cumulative literature should be considered as a whole—ideally, some studies will feature the same limitations, but others will have addressed them.

This is not to say that eyewitness science has a perfect record. Some studies have been undermined by subsequent findings, resulting in updates to policy recommendations. Some subtopics remain underdeveloped and still lack the ability to comprehensively explain the reasons for and meaning of particular eyewitness behaviors and decisions. Applying the current body of research in practice is also challenging due to ethical and practical difficulties associated with obtaining data from real eyewitnesses in real cases. Nevertheless, there is a substantial and consistent body of evidence showing that eyewitness accuracy improves when police use a double-blind, fairly constructed lineup with clear, unbiased instructions—enough to support strong confidence in the real-world benefits of these practices ([73]).

Before LPE can reach a comparable level of foundational validity, the field must develop a similarly systematic and reproducible evidence base. Moreover, we argue that researchers in the LPE discipline move away from relying on large-scale, black-box studies. Though such studies are critical, a range of high-quality empirical work would contribute meaningfully to this literature, such as controlled experiments, field research, and mixed methods approaches that examine how specific procedures and contextual factors influence examiner performance. No single study can answer every question about how easy it is to use and adhere to a forensic method, or the associated performance of that method—but a diverse, cumulative body of research can gradually build a coherent understanding, despite the limitations of individual projects.

### 4.2. Latent Fingerprint Examination Does Not Have a Standardized Methodology

Perhaps the most important issue relevant to the foundational validity of LPE is the absence of a standardized methodology that is used in practice. While studies show that LPEs can be accurate when comparing latent and exemplar prints, the field is missing guidance on the circumstances, procedures, and decision thresholds that guarantee those levels of accuracy. To be clear, the goal is not to eliminate professional discretion, but the amount of leeway given to examiners in their work should be intentional and, ideally, based on empirical research.

Without first determining how LPEs should approach examinations given evidence of different levels of quality and different case circumstances, assessing the reliability of the discipline is impossible. In contrast, eyewitness research has long focused on two core procedures—the showup (presentation of a single suspect) and the lineup (presentation of a suspect embedded among fillers known to be innocent). Both procedures have been evaluated extensively under different conditions, enabling researchers to offer empirically grounded recommendations about how specific procedural choices influence outcomes—such as lineup size, instructions, or selection criteria.

Foundational validity, by definition, requires a defined method and evidence of how that method performs under known conditions. The lack of standardization in LPE creates a serious barrier to advancing latent fingerprint examination along the foundational validity continuum. In the United States, LPE practices vary widely across jurisdictions, FSPs, and even individual examiners, with not universally applied decision thresholds or procedures. Although multiple validated methods could be acceptable, standardization should be preceded by empirical testing of potential approaches to ensure they support inter- and intra-examiner consistency. Without, defined, consistently applied methods, estimating the potential for error in any particular case is scientifically indefensible. Importantly, this challenge is not unique to LPE—it reflects a broader problem across many forensic science disciplines.

### 4.3. Lessons from Eyewitness Reform and the Path Forward for Latent Prints

Eyewitness errors remain the most common contributing factor to wrongful convictions in DNA exoneration cases, and data suggests that these errors have contributed to more than 60% of Innocence Project exonerations ([26]). Eyewitness evidence is especially influential—both during investigations and in court proceedings ([17]). That said, misapplied or invalidated forensic science, including latent print evidence, follows closely behind, contributing to roughly half of all DNA exoneration cases ([26]). This is especially troubling in light of other evidence that laypeople ([6]; [13]) and legal professionals ([10]) tend to struggle with evaluating the reliability and probative value of forensic evidence. Given the real-world consequences of unreliable evidence, a clear understanding of the circumstances under which these forms of evidence can be relied upon is critical to safeguarding the integrity of the justice system.

When the Innocence Project began using DNA to exonerate wrongfully convicted individuals and identify sources of error, the eyewitness literature was already equipped with a strong empirical foundation ([78]) and expert consensus ([28], [29]). As a result, when reforms were needed, the field was prepared to offer concrete, evidence-based recommendations. In contrast, other forensic disciplines—particularly the pattern-matching sciences—were not similarly prepared, a fact underscored by the findings in the 2009 National Research Council report.

This does not mean that LPE decision-making cannot achieve a level of scientific rigor comparable to eyewitness identification. The field has already shown potential, with early research providing important insights. These accomplishments could be effectively highlighted through large-scale, critical reviews, similar to the two eyewitness SRPs discussed here, to synthesize what is well established and identify areas needing further study. What is concerning, however, is the growing inclination to treat a small set of studies as sufficient to establish foundational validity, or to dismiss targeted, well-designed studies simply because they do not meet PCAST’s strict criteria for black-box research. Furthermore, LPE—and forensic science disciplines generally—would benefit from reconceptualizing foundational validity as a continuum rather than a fixed destination ([80]). To advance, the field must commit to ongoing programmatic research efforts, the development and validation of standardized methods, and a commitment to identifying the features of methods that facilitate accurate LPE judgments—and those that do not.

## Figures and Tables

**Figure 1 behavsci-15-01145-f001:**
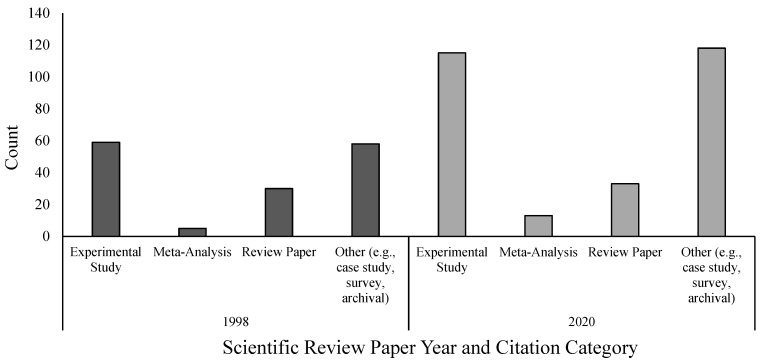
Types of citations in the 1998 and 2020 Scientific Review Papers on eyewitness identification science.

**Figure 2 behavsci-15-01145-f002:**
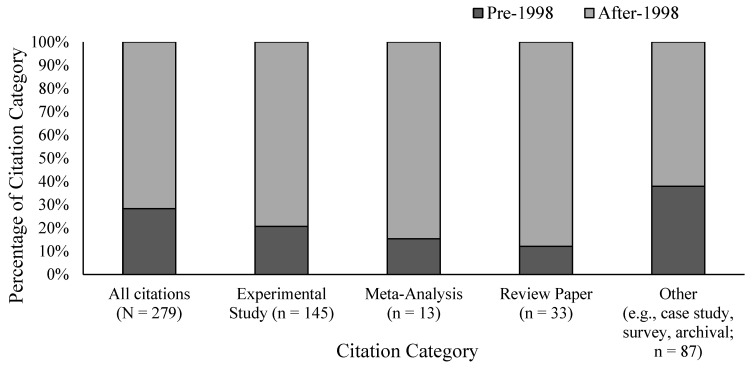
Citations in the 2020 Scientific Review Paper on eyewitness identification science published pre- and post-1998.

**Table 1 behavsci-15-01145-t001:** Comparison of the different elements of eyewitness identification procedures and latent fingerprint examination and the use of that evidence.

Elements of Procedures	Eyewitness Identification	Latent Print Examination
Crime Scene Data?	Memory of people present during or around the time of a crime event.	Latent fingerprint collected in connection with a crime.
Recommended Method?	Fair, double-blind lineup procedure with unbiased instructions.	ACE-V and any local standard operating procedures.
Expertise Required?	Yes—but laypersons are a type of face recognition/matching “expert”.	Yes—special training and experience required.
Goal?	Link a particular person to a crime event or criminal act.	Link a particular person to a crime event or criminal act.
End-Users?	Police, lawyers, people accused of crimes, judges, and jurors.	Police, lawyers, people accused of crimes, judges, and jurors.

## Data Availability

Data were derived from coding the reference lists in [78] ([78], [73]). Our coded data can be accessed on OSF (https://osf.io/68srz/ (accessed on 15 June 2025)).

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
