# Peer review of "On the Continuum of Foundational Validity: Lessons from Eyewitness Science for Latent Fingerprint Examination"

_behavsci, 2025, doi:10.3390/bs15091145_

Round 1

Reviewer 1 Report

Comments and Suggestions for Authors

Review of On the Continuum of Foundational Validity

This manuscript nicely contrasts the literature on eyewitness testimony against the literature on latent print examinations. This is an important comparison that makes critical comparisons that are designed to spur both fields forward. I suspect that this will be acceptable for publication with minor changes.

The authors review both literatures, and then conclude that fingerprints have lower foundational validity than eyewitness identification. This will come as a surprise to most readers, because introductory psychology classes for a long time have taught that eyewitness testimony is unreliable (with reasonable support for that claim). Thus it will be important for the authors to distinguish accuracy from validity. To that end, it might be helpful to clarify those dimensions or qualities along which we judge foundational validity.

Counting publications seems like a crude metric. Although journals like Journal of Forensic Identification are not indexed, they contain many studies that are of the type cited for eyewitness identification. Perhaps the authors reference these citations to prod the fingerprint field to write more Scientific Review Papers.

I’m of the opinion that the eyewitness testimony literature is summarized with an optimistic tone. For example, the cumulative work of perhaps hundreds of papers settled on the new recommendation of successive lineups in contrary to the simultaneous lineups. However, as far as I know, this effort was quietly rolled back after the field discovered that this was not some new way to improve accuracy or increase justice, but merely lead to a criterion shift on the part of subjects. Whether this is good or bad depends on whether you are the prosecutor or the defense attorney. Any field will have its missteps, but this was a major recommendation that was trumpeted as a great success and then retracted. This in my view serves as a warning light for recommended changes in other fields, and doesn’t speak highly of the eyewitness field as a whole. I suppose one could view this as an example of how one establishes foundational validity, but it seems embarrassing at a minimum.

The article goes over error rates for fingerprints, but doesn’t review the literature on eyewitness identification error rates. Is there a reason for this? Should accuracy even be part of the equation for establishing foundational validity?  

I don’t like this sentence: “Moreover, existing error rate studies tend to rely on fingerprint impressions of relatively high quality, which does not reflect the typical quality encountered in real-world casework.” Define real-world casework. Please. Every single paper that criticizes error rate studies or proficiency tests seems to say that the studies are too easy or to hard relative to casework, but they never seem to say what casework is. I realize you are citing others, but if they can’t define it either, then don’t repeat their sloppy scholarship. Or perhaps you can define casework and justify their claims.

I think a related question for the eyewitness literature is whether the studies are remotely similar to “casework”. How can the tedium of the inside of a psychology building laboratory or the safe environment of a Prolific experiment correspond to the terror of an active crime? It seems like foundational validity should consider the context of the testing relative to the real-world analog of each discipline. If most eyewitness studies lack the emotional impact of crime-related memories, are they even applicable? Perhaps some authors have tried to address this, although modern IRBs will make this very difficult. This disconnect should bear on the question of foundational validity of eyewitness testimony.

In sum, I would characterize the eyewitness literature as wide but shallow given the challenges of replicating the emotional impact of crime scenes. Fingerprints has a smaller literature but benefits from (albeit imperfect) professionalization among the practitioners. Although I will not defend ACE-V, there are many attempts at deriving an “alphabet” of features and process maps that attempt to explain the mental processes that underlie a conclusion. I can think of no analog in the eyewitness literature. Such efforts should contribute to foundational validity. The decision threshold is poorly defined, but verification and the ability for others to access the same evidence tend to offset differences in thresholds.

Other comments:

What is the evidence that supports this statement “Eyewitness identification evidence is frequently used in the criminal legal system and is more foundationally valid than almost every forensic discipline…” The previous paragraph labeled fingerprints as foundationally valid.

Author Response

1.1 This manuscript nicely contrasts the literature on eyewitness testimony against the literature on latent print examinations. This is an important comparison that makes critical comparisons that are designed to spur both fields forward. I suspect that this will be acceptable for publication with minor changes.

We thank Reviewer #1 for their thoughtful comments and are pleased that their reaction was positive! We believe that the revisions inspired by these suggestions have improved the manuscript substantially.

1.2 The authors review both literatures, and then conclude that fingerprints have lower foundational validity than eyewitness identification. This will come as a surprise to most readers, because introductory psychology classes for a long time have taught that eyewitness testimony is unreliable (with reasonable support for that claim). Thus it will be important for the authors to distinguish accuracy from validity. To that end, it might be helpful to clarify those dimensions or qualities along which we judge foundational validity.

Reviewer #1 makes an important point about clearly distinguishing between what it means to have an accurate discipline, or know how accurate a discipline is, and whether a discipline has foundational validity. A discipline can be very inaccurate, but foundationally valid, and vice versa. We also agree that it would be helpful to provide some suggestions about elements of a discipline or literature that researchers and practitioners could focus on when assessing foundational validity. So, we have now made some changes to ensure the definition of foundational validity (and how it is distinct from accuracy) is clear, as well as providing some dimensions upon which foundational validity could be judged. We believe that this clarifies PCAST’s position as well as our own points about methodological precision and why establishing foundational validity can require a variety of empirical approaches.

On page 2, lines 54 to 60, we now clarify the way in which PCAST conceptualized foundational validity in forensic science by stating:

“Their mandate was to determine whether these disciplines had demonstrated foundational validity, defined as the extent to which a method has been empirically shown to produce accurate and consistent results, based on peer-reviewed, published studies (PCAST, 2016). Specifically, PCAST evaluated whether each discipline’s procedures had been tested for repeatability (within examiner), reproducibility (across examiners), and accuracy under conditions representative of actual casework (see also the discussion in Stern et al., 2019).”

We also clearly state that foundational validity is a feature of the method itself, rather than to how that method performs, on page 2, lines 61 to 71:

“Importantly, PCAST emphasized that foundational validity is a property of the specific method under consideration, rather than a property of performance outcomes. That is, a discipline can lack foundational validity even when examiners achieve accurate results provided that their success cannot be not attributed to a clearly defined and consistently applied method that can be independently replicated (see also Wilson et al., 2020). Given these implications, it is unsurprising that methodological precision and compliance have received increasing attention in recent years (Swofford et al., 2024). Without a clearly specified and consistently applied method, results from studies designed to observe performance metrics reflect the accuracy achieved by an undefined mix of examiner strategies that cannot be meaningfully linked to any particular approach and are, consequently, difficult to interpret, predict, or replicate.”

1.3 Counting publications seems like a crude metric. Although journals like Journal of Forensic Identification are not indexed, they contain many studies that are of the type cited for eyewitness identification. Perhaps the authors reference these citations to prod the fingerprint field to write more Scientific Review Papers.

We agree that counting publications is an imprecise measure of literature quality. However, for the eyewitness field, our counts were of studies, reviews, and theory papers selected by experts in the field, so these are all—at least to some degree—considered high impact in the field for the purpose of making research-based recommendations to police agencies. We now make this point on page 5, lines 217 to 221:

“Of course, the raw number of studies cited in a review paper is a crude measure of scientific progress. However, these citations were selected by experts to support formal policy recommendations and vetted through several rounds of comments, suggesting they meet a standard of quality broadly accepted by researchers in the field.”

Although we provide no counts of LPE studies, we include explicit references to a wider range of fingerprint studies than were discussed in previous version of this paper, both when discussing the PCAST report (page 2, lines 72 to 81) and in the LPE focused section of the paper (Section 3, page 8 lines 301 to 316).

1.4 I’m of the opinion that the eyewitness testimony literature is summarized with an optimistic tone. For example, the cumulative work of perhaps hundreds of papers settled on the new recommendation of successive lineups in contrary to the simultaneous lineups. However, as far as I know, this effort was quietly rolled back after the field discovered that this was not some new way to improve accuracy or increase justice, but merely lead to a criterion shift on the part of subjects. Whether this is good or bad depends on whether you are the prosecutor or the defense attorney. Any field will have its missteps, but this was a major recommendation that was trumpeted as a great success and then retracted. This in my view serves as a warning light for recommended changes in other fields, and doesn’t speak highly of the eyewitness field as a whole. I suppose one could view this as an example of how one establishes foundational validity, but it seems embarrassing at a minimum.

Reviewer #1 makes a good point about some of the “missteps” in eyewitness identification—and we think that these help to make an important point in this paper too. There have been several policy recommendations that were made in the past but, in light of new evidence, no longer appear in the recent Scientific Review Paper (2020). We now explicitly discuss some of these examples after introducing the 1998 and 2020 SRPs, beginning on page 6, line 227. Here, there are now two paragraphs discussing some issues with specific estimator and system variables that have come to light over time in eyewitness science. These additions are important, we think, as they are excellent examples of how cumulative science can and should be self-correcting—progress like this is not inherently bad, though it can be unfortunate in applied fields if recommendations and policy change have already happened.

1.5 The article goes over error rates for fingerprints, but doesn’t review the literature on eyewitness identification error rates. Is there a reason for this? Should accuracy even be part of the equation for establishing foundational validity?  

We now clearly separate accuracy (a performance metric of a method) from foundational validity (a feature or property of a method) but also discuss eyewitness identification accuracy more clearly to illustrate the difference between these concepts. See page 3, lines 117 to 124:

“Yet, eyewitnesses are mistaken at an alarming rate. Even when identification procedures follow best practices, approximately one-third of real eyewitnesses identify a known-innocent person (an intentionally placed filler: Wells et al., 2015). So, although research shows that eyewitnesses are less accurate than LPEs, there is a robust body of empirical research supporting the methods recommended for use in practice, which is important for foundational validity. This makes eyewitness identification a compelling example of how foundational validity can be achieved even in the face of known performance limitations.”

1.6 I don’t like this sentence: “Moreover, existing error rate studies tend to rely on fingerprint impressions of relatively high quality, which does not reflect the typical quality encountered in real-world casework.” Define real-world casework. Please. Every single paper that criticizes error rate studies or proficiency tests seems to say that the studies are too easy or to hard relative to casework, but they never seem to say what casework is. I realize you are citing others, but if they can’t define it either, then don’t repeat their sloppy scholarship. Or perhaps you can define casework and justify their claims.

We now clarify the point that we were hoping to make here. Other researchers have critiqued the existing error rate studies because they do not “reflect the typical quality encountered in real-world casework” but have been frustratingly vague about what that means (as Reviewer #1 points out). To begin with, it is difficult to accurately represent case work because that will vary by jurisdiction, lab, and type of case. What base rate should be used for same- and different-source judgments if we wish to reflect real casework? How many should be poor, moderate, and high quality? What does a high, moderate, and low-quality latent fingerprint look like in casework? These are all questions that would need to be answered to even being to create a study that accurately represents real casework.

This issue further supports the cumulative science point at the heart of this paper. It is practically impossible to create a set of materials and testing conditions that mirrors the range of conditions that examiners are likely to experience when completing casework. Thus, to begin to understand how casework impact LPE, experimental scientists might pick an important factor in casework or a dimension upon which casework varies and attempt to understand that condition (or several of those conditions), as well as how they impact consistency and accuracy among examiners.

For instance, Koehler & Liu (2021) focused on the issue of close non-matches—how does consistency between examiner and accuracy vary when latent fingerprints that are close non-matches are included in the study materials? Another paper by Growns and Kukucka (2021) examined the effect of a low, equal, or high prevalence of same and different source pairs on naive participant’s source judgments. Both of these papers have various limitations but also contribute to the literature in important ways by shedding light on how LPE decisions are likely to be made under specific conditions.

We now include a bit more of a discussion of this on page 9, beginning line 380, though we do not try to tackle this topic in detail—that could be a paper on its own, probably. Here is an excerpt from the new manuscript:

“Moreover, existing error rate studies tend to rely on fingerprint impressions that are already processed to some degree, are higher quality than latent fingerprints lifted from crime scenes, unknown frequencies of same- and -different source latent-exemplar pairs in regular casework, and examined under conditions where the LPEs know that they are being tested by researchers who will compare their judgments to the correct answers. These are important deviations from casework that are known to impact how people make decisions—generally and specifically in LPE (Growns & Kukucka, 2021; Haber & Haber, 2014; Scurich et al., 2025; see also critiques of fingerprint proficiency testing in Kelley et al., 2020; Koertner & Swofford, 2018). Thus, the error rates in operational settings may be different than those reported in the literature—particularly when considering the prevalence of close non-matches (Koehler & Liu, 2021) and the influence of task-irrelevant, biasing contextual information (Dror & Kukucka, 2021; Quigley-McBride et al., 2022) in real cases which tend to change decision-making patterns.”

1.7 I think a related question for the eyewitness literature is whether the studies are remotely similar to “casework”. How can the tedium of the inside of a psychology building laboratory or the safe environment of a Prolific experiment correspond to the terror of an active crime? It seems like foundational validity should consider the context of the testing relative to the real-world analog of each discipline. If most eyewitness studies lack the emotional impact of crime-related memories, are they even applicable? Perhaps some authors have tried to address this, although modern IRBs will make this very difficult. This disconnect should bear on the question of foundational validity of eyewitness testimony.

There was a brief discussion of this issue in the original manuscript, but we have not expanded that discussion. This appears page 7, beginning line 265, and addresses the field studies conducted in eyewitness identification that are limited, but useful for demonstrating that the laboratory effects extend to real cases.

1.8 In sum, I would characterize the eyewitness literature as wide but shallow given the challenges of replicating the emotional impact of crime scenes. Fingerprints has a smaller literature but benefits from (albeit imperfect) professionalization among the practitioners. Although I will not defend ACE-V, there are many attempts at deriving an “alphabet” of features and process maps that attempt to explain the mental processes that underlie a conclusion. I can think of no analog in the eyewitness literature. Such efforts should contribute to foundational validity. The decision threshold is poorly defined, but verification and the ability for others to access the same evidence tend to offset differences in thresholds.

We believe that by addressing the comments provided by Reviewer #1, each in turn, we have strengthened the manuscript in such a way that this summary of concerns has been addressed. If there are any remaining suggestions or concerns, we would be happy to consider them and incorporate those too, where appropriate.

1.9 Other comments: What is the evidence that supports this statement “Eyewitness identification evidence is frequently used in the criminal legal system and is more foundationally valid than almost every forensic discipline…” The previous paragraph labeled fingerprints as foundationally valid.

We have edited this opening sentence of Section 2 so that it now reads, “Eyewitness identification evidence is frequently used in the criminal legal system and has a broad body of research guiding best practices. Of course, this was not always the case.” (see page 4, lines 169 to 171).

Reviewer 2 Report

Comments and Suggestions for Authors

The paper is well written and intuitively compelling. However, my main concern lies with the concept of a "black-box study" as it is applied to latent print examination (LPE) research, but not with equivalent precision to eyewitness memory research. The term “black-box” appears only four times in the manuscript, and its use is not clearly defined or consistently applied. Given its centrality to the authors’ argument about the limitations of the LPE literature, the concept either needs to be more rigorously developed or potentially removed altogether.

If the authors intend to retain the black-box framing, I outline several concerns below regarding how it currently creates a somewhat misleading dichotomy between LPE and eyewitness research. In the conclusion, the authors themselves suggest that black-box criteria may be impeding progress in LPE, which is an important point. If so, they might consider more clearly articulating how eyewitness research benefited from not being held to those same criteria. That contrast could be a useful contribution in its own right.

A central claim of the manuscript is that LPE lacks foundational validity in part because only three black-box studies exist, whereas eyewitness identification is supported by a broader empirical base. However, the criteria for what qualifies as a black-box study are never clearly defined. Without a precise, operational definition, it is unclear why so many studies in the LPE literature (many of which meet core features such as blinded conditions, known ground truth, and outcome-based performance metrics) are excluded from the black-box category. Simultaneously, it is not specified which eyewitness studies are being counted as black-box analogs or why they qualify.

If the authors are using a more restrictive definition (e.g., requiring both target-present and target-absent trials, or specific field-based validation), that should be stated explicitly and applied consistently across both domains. Otherwise, the current framing risks imposing a higher evidentiary bar on LPE research than on eyewitness research, introducing a perceived asymmetry that undermines the comparative claims. It may well be true that more eyewitness studies meet these criteria, but in its current form, the argument comes across as imprecise and, at times, hand-wavy.

I would strongly recommend that the authors (1) clearly define what they mean by a black-box study, and (2) systematically evaluate which studies in both literatures meet that definition. If the core thesis is instead that eyewitness science has benefited from decades of sustained research while LPE is relatively newer and less mature, that framing may be more appropriate and defensible than relying on the under-defined black-box distinction. As currently written, the argument is intuitive but lacks the clarity and rigor needed to be fully persuasive.

Finally, the authors’ conceptualization of foundational validity as a continuum is a valuable and progressive framing. It closely parallels the argument made by Wilson, Harris, and Wixted (2020, PNAS), which contends that scientific validity should not be treated as a binary (“signal” or “noise”) but rather as a continuous variable. The authors might consider drawing on that work, as it could strengthen the theoretical grounding of their claim that foundational validity is not a yes-or-no threshold, but an evolving state of evidence accumulation.

This is a well-written and important manuscript that makes a valuable contribution to discussions around foundational validity in forensic science. However, I recommend the authors clarify and substantiate one of the paper’s central claims: the comparison between latent fingerprint examination and eyewitness memory research based on the notion of “black-box” studies. As currently presented, the argument relies on an under-defined and asymmetrically applied standard that risks obscuring rather than illuminating differences between the two domains. With a more precise definition of terms, a consistent application of criteria across both literatures, and a clearer articulation of whether the key issue is research methodology or research maturity, the manuscript will be substantially strengthened and better positioned to support its important thesis.

Author Response

2.1 The paper is well written and intuitively compelling. However, my main concern lies with the concept of a "black-box study" as it is applied to latent print examination (LPE) research, but not with equivalent precision to eyewitness memory research. The term “black-box” appears only four times in the manuscript, and its use is not clearly defined or consistently applied. Given its centrality to the authors’ argument about the limitations of the LPE literature, the concept either needs to be more rigorously developed or potentially removed altogether.

We would like to thank Reviewer #2 for highlighting an issue with the clarity of our argument in the original paper. Though the black-box studies in LPE are essential to our points in this paper, we believe that we had not made it clear in what way these studies needed to be discussed here: instead of them being provided as an example of the type of work that we think there should be more of, we recognize their importance while encouraging a much broader range of research. We hope that is clear in the revised version of the manuscript.

We have now included a definition of black-box study when the term first appears in a footnote. This includes a general definition, as well as the specific criteria applied by PCAST in their 2016 report. It appears on page 2 and reads:

“A test of forensic examiner performance only—the decision processes and reasoning of the examiners is not evaluated, just the final decisions (e.g., value/no value judgments and identification/inconclusive/exclusion decisions). PCAST stated that they were studies in which “many examiners are given the same test items and asked to reach conclusions, without information about the ground truth or about the responses of other examiners. The study measures how often examiners reach correct conclusions, erroneous conclusions, and inconclusive results.” (PCAST, 2016, p. 6) They also specified that, in addition to measuring examiner performance, the materials and testing should resemble casework, and the participants should complete an examination from start to finish (as they would in a real case).”

Black-box studies are less common in psychology unless a research area is very new. There are some subdisciplines within psychology that may use them--such as behaviorism or some applied fields (e.g., plea bargaining in psychological literature is conceived of as a black-box, but the black box are various people's cognitions operating within a legal system that is ill-defined rather than an individual's cognitive processes). Black-box studies do not, however, shed light on the psychology of decision-making and judgments among eyewitnesses and LPEs--research in that area is very interested in how the judgments and decisions are made, not just what judgment or decision is made. So most psychologists would not find it as interesting to run such a study unless the manipulated variables shed light on something relevant to psychological theory or processes. The reasons why people make particular judgments and decisions is the focus of psychological research. Experimental psychologists will seek to determine why an effect occurs, as well as the boundary conditions of that effect (and the reasons those are the boundary conditions). So, there may be some studies out there that are similar to black-box studies in forensic science and other applied fields, but they are not conceptualized that way in psychology or prioritized, which is the reason that phrase did not appear in the discussions of eyewitness identification.

2.2 If the authors intend to retain the black-box framing, I outline several concerns below regarding how it currently creates a somewhat misleading dichotomy between LPE and eyewitness research. In the conclusion, the authors themselves suggest that black-box criteria may be impeding progress in LPE, which is an important point. If so, they might consider more clearly articulating how eyewitness research benefited from not being held to those same criteria. That contrast could be a useful contribution in its own right. 

We agree and in the new manuscript we have attempted to make it clearer that we believe that LPE would benefit from embracing a range of empirical approaches rather than focusing on a very specific type of study that yields answers to only a fairly narrow set of research questions.

2.3 A central claim of the manuscript is that LPE lacks foundational validity in part because only three black-box studies exist, whereas eyewitness identification is supported by a broader empirical base. However, the criteria for what qualifies as a black-box study are never clearly defined. Without a precise, operational definition, it is unclear why so many studies in the LPE literature (many of which meet core features such as blinded conditions, known ground truth, and outcome-based performance metrics) are excluded from the black-box category. Simultaneously, it is not specified which eyewitness studies are being counted as black-box analogs or why they qualify.

This was actually a central point that we wished to make—black-box studies serve a specific and important purpose in some fields, but they are not the only type of research that contribute to scientific understanding and foundational validity. We disagree with PCAST’s assessment that very specific types of black-box studies, and only those studies, are essential to establishing foundational validity. We have tried to make this clearer throughout the paper so that the reader easily picks up on this point. For example, page 2 beginning line 82:

“So, is the existing body of empirical research on LPE sufficient to establish its foundational validity? In experimental psychology, most researchers would likely agree that three studies conducted under a narrow set of conditions may be promising and worthy of further exploration, but insufficient for making broad policy recommendations about the practices being evaluated. At the same time, researchers in psychology would not necessarily dismiss studies that fall outside of the strict black-box criteria outlined by PCAST (2016); a diverse range of empirical work can offer valuable insights.”

And page 9, beginning line 341:

“However, this conclusion was based on “only two properly designed studies of the foundational validity and accuracy of latent fingerprint analysis” (p.101), as other studies measuring error rates among LPEs failed to meet PCAST’s criteria for methodological rigor. Although it is standard in science to interpret results cautiously based on study limitations, the studies dismissed by PCAST are not without value. In fact, subsequent peer-reviewed publications referring to the report have critiqued PCAST’s treatment of the LPE literature and the question of foundational validity (e.g., Hicklin et al., 2025; Koehler & Liu, 2021).”

2.4 If the authors are using a more restrictive definition (e.g., requiring both target-present and target-absent trials, or specific field-based validation), that should be stated explicitly and applied consistently across both domains. Otherwise, the current framing risks imposing a higher evidentiary bar on LPE research than on eyewitness research, introducing a perceived asymmetry that undermines the comparative claims. It may well be true that more eyewitness studies meet these criteria, but in its current form, the argument comes across as imprecise and, at times, hand-wavy.

We are not using that definition, but we do now make that explicit—see our response to 2.3 above.

2.5 I would strongly recommend that the authors (1) clearly define what they mean by a black-box study, and (2) systematically evaluate which studies in both literatures meet that definition. If the core thesis is instead that eyewitness science has benefited from decades of sustained research while LPE is relatively newer and less mature, that framing may be more appropriate and defensible than relying on the under-defined black-box distinction. As currently written, the argument is intuitive but lacks the clarity and rigor needed to be fully persuasive.

We do clearly define “black-box study” now, but we have not made changes to address the second point here. This is because this was not the goal of the paper and, given that black-box studies are not a focus in psychological literature, this wouldn’t be a productive exercise when assessing eyewitness identification (see our response to 2.1 above). Our point is not to encourage more black-box studies (though we agree that they are important and serve a purpose for foundational science in applied settings), but rather to encourage a broader range of research specifically designed to address the way people/experts approach LPE, unifying the varied approaches under one or several clear, well-defined procedures so that performance and conformance can be empirically assessed to confirm the level of performance that those approaches can be expected to produce, and when those approaches do and do not work well.

2.6 Finally, the authors’ conceptualization of foundational validity as a continuum is a valuable and progressive framing. It closely parallels the argument made by Wilson, Harris, and Wixted (2020, PNAS), which contends that scientific validity should not be treated as a binary (“signal” or “noise”) but rather as a continuous variable. The authors might consider drawing on that work, as it could strengthen the theoretical grounding of their claim that foundational validity is not a yes-or-no threshold, but an evolving state of evidence accumulation.

We would like to thank Reviewer #2 for bringing this paper to our attention. We have read the paper and incorporated it into the manuscript (see page 2, lines 62 to 65 and page 13, lines 551 to 553).

2.7 This is a well-written and important manuscript that makes a valuable contribution to discussions around foundational validity in forensic science. However, I recommend the authors clarify and substantiate one of the paper’s central claims: the comparison between latent fingerprint examination and eyewitness memory research based on the notion of “black-box” studies. As currently presented, the argument relies on an under-defined and asymmetrically applied standard that risks obscuring rather than illuminating differences between the two domains. With a more precise definition of terms, a consistent application of criteria across both literatures, and a clearer articulation of whether the key issue is research methodology or research maturity, the manuscript will be substantially strengthened and better positioned to support its important thesis.

We would like to thank Reviewer #2 for their useful recommendations that we believe have strengthened the manuscript and helped to clarify our key points for readers. In particular, we hope that our point about black-box studies is much clearer now--that they are not the only source of valuable data for the purpose of establishing a scientific foundation in a discipline.